# The Role of Extracellular Polymeric Substances in the Toxicity Response of Anaerobic Granule Sludge to Different Metal Oxide Nanoparticles

**DOI:** 10.3390/ijerph19095371

**Published:** 2022-04-28

**Authors:** Huiting Li, Fang Chang, Zhendong Li, Fuyi Cui

**Affiliations:** 1Tianjin Research Institute for Water Transport Engineering, M. O. T, Tianjin 300000, China; ophelia_lee@163.com (H.L.); changfang1999@163.com (F.C.); lizhendong0517@163.com (Z.L.); 2College of Environment and Ecology, Chongqing University, Chongqing 400045, China

**Keywords:** metal oxide nanoparticles (NPs), extracellular polymeric substances (EPSs), anaerobic granular sludge (AGS), population dynamics, toxicity mechanisms, interactions, anaerobic digestion

## Abstract

Wastewater treatment plants (WWTP) are regarded as the last barriers for the release of incompletely separated and recycled nanoparticles (NPs) into the environment. Despite the importance and ubiquity of microbial extracellular polymeric substances (EPSs) in the complex wastewater matrix, the interaction between NPs and EPSs of anaerobic microflora involved in wastewater treatment and the resultant impact on the biomass metabolomics are unclear. Thus, the impacts of different metal oxide (TiO_2_, ZnO, and CuO) NPs on functional bacteria in anaerobic granular sludge (AGS) and the possible toxicity mechanisms were investigated. In particular, the binding quality, enhanced resistance mechanism, and chemical fractional contribution of EPSs from AGS against the nanotoxicity of different NPs was assessed. The results showed that CuO NPs caused the most severe inhibition towards acetoclastic and hydrogenotrophic methanogens, followed by ZnO NPs, whereas TiO_2_ NPs caused no inhibition to methanogenesis. Excessive EPS production, especially the protein-like substances, was an effective strategy for reducing certain NPs’ toxicity by immobilizing NPs away from AGS cells, whereas the metabolism restriction on inner microorganisms of AGS induced by CuO NPs can deteriorate the protective role of EPS, indicating that the roles of EPS may not be amenable to generalizations. Further investigations with lactate dehydrogenase (LDH) and reactive oxygen species (ROS) assays indicated that there are greatly essential differences between the toxicity mechanisms of metal NPs to AGS, which varied depending on the NPs’ type and dosage. In addition, dynamic changes in the responses of EPS content to different NPs can result in a significant shift in methanogenic and acidogenic microbial communities. Thus, the production and composition of EPSs will be a key factor in determining the fate and potential effect of NPs in the complex biological matrix. In conclusion, this study broadens the understanding of the inhibition mechanisms of metal oxide NPs on the AGS process, and the influence of EPSs on the fate, behavior, and toxicity of NPs.

## 1. Introduction

Due to their variations in size, shape, composition, and origin, metal oxide nanoparticles (NPs) offer a greater surface area than their bulk counterparts, allowing for improved performance in both industrial and household use, but hold even greater promise for future applications. As a result, concerns have arisen about the presence of these NPs in the environment and the potential threat for human and environmental health. Recently, many researchers have focused on the absorption and toxicity of metal oxide NPs on different organisms, and especially the antibacterial action of metal oxide NPs, since bacteria act as decomposers and play an important role in maintaining the ecosystem in natural and engineered systems. For example, the antibacterial properties towards lactic acid bacteria [1] have been reported in a study comparing the toxicity of ZnO and Ag NPs towards mesophilic and halophilic bacterial cells, in which ZnO NPs were found to be more potent regarding cytotoxicity on growth and viability. A study of the sub-toxic effects of CuO NPs on *Escherichia coli* indicated that CuO NPs can induce a single-stranded DNA break at 0.1 mg Cu/L [2].

The occurrence of metal oxide NPs in wastewater and sewage sludge has been detected in wastewater treatment plants (WWTPs), which are regarded as the last barriers for the release of NPs into the environment [3]. Since the absorption, aggregation, and settling by activated sludge is considered to be the primary physical removal mechanisms of these NPs [4], there is an urgent need to explore the impact of metal oxide NPs on the biomass involved in wastewater treatment. Although recent studies have focused on the effect of metal oxide NPs on the activated sludge process and some instructive results have been obtained, emphasis has been placed on the microorganisms of traditional activated sludge [5] or waste sludge fermentation [6], which may not be appropriate to interpret the fate and influence of NPs on the microbial communities mediating the anaerobic treatment of NP-containing wastewater. Few studies have investigated the toxicity of metal oxide NPs for anaerobic wastewater treatment systems, and different possible explanations have been proposed [7,8]. Nevertheless, there remains a lack of information regarding a systematic comparison among the responses of anaerobic microflora toward different metal oxide NP types in the complex wastewater matrix.

Due to its high biomass concentration, dense and strong microbial aggregate structure, and excellent settling capacity, anaerobic granular sludge (AGS) has been proven to have excellent properties for the treatment of various types of industrial wastewater, including cosmetics, pharmaceutics, and textile wastewater, which are the major sources of the release of metal oxide NPs [9]. The AGS can be divided into three zones, i.e., a central core of acetoclastic methanogens is surrounded by a layer of acetogens and hydrogenotrophic methanogens, and by an outside layer of acidogens that hydrolyze and acidify complex organic matter [10]. Therefore, the response of AGS to metal oxide NPs may be different to that of activated sludge because of its unique granular attributes. However, little information is currently available regarding this point, and especially regarding the toxicity of metal oxide NPs for the methanogenic activity of acetoclastic and hydrogenotrophic methanogens, because these two population changes can reflect the stability of an anaerobic treatment system and the balance of actions and interactions of complex microbial communities [11].

Extracellular polymeric substances (EPSs) are defined as complex high-molecular-weight mixtures of polymers, which can form a gel-like network to keep bacteria together, cause the adherence of biofilms to surfaces, and protect bacteria against noxious environmental conditions [12]. Due to the presence of charged functional groups, EPSs ideally serve as a natural ligand source, providing for other charged particles or molecules including metal oxide NPs, which may be one of the determinants of the fate of NPs in anaerobic wastewater treatment system [8]. For AGS, most of the EPSs are distributed in the outer layer, which means that EPS are usually the first barrier of AGS microbial cells that directly contacts and interacts with the toxic NPs, and may reduce their potential toxicity to functional bacteria in AGS [13]. Although certain researchers have noted that the characteristics of EPSs are obviously influenced by metal oxide NPs in aerobic activated sludge, few studies have been conducted to investigate the role of EPSs in AGS under metal oxide NPs’ exposure; however, the results are inconsistent, showing that EPS content and NP dosage are positively [14] or negatively [15] correlated. Given that different seed AGS sources and medium compositions were adopted in these studies, comparing the above results seems to be quite difficult due to the lack of a uniform test criterion. Therefore, the exact roles of EPSs in response to NP-induced toxicity to AGS still need to be identified. In addition, more studies are required to systematically understand which components of EPSs affect the fate and the toxicity of different metal oxide NPs under anaerobic conditions, and their means of doing so. This help to provide a comprehensive understanding of the risk of NPs on the activated sludge process and effectively utilize the advantages of metal oxide NPs.

Based on the above discussion, the objective of this study was to evaluate and compare the potential toxicity of metal oxide (TiO_2_, ZnO, and CuO) NPs to the AGS methanogenic activity and the respective roles of EPSs under different NPs’ exposure. Specifically, this study focused on the population dynamics of anaerobic digestion processes and how they are linked to the properties and composition of the EPS when different types of NPs were released to wastewater.

The effects of metal oxide NPs and their corresponding released ions on the activity of methanogens were investigated. Next, the composition of the EPS and the consequent alterations of population dynamics in anaerobic digestion processes after exposure to different metal oxide NPs were carefully studied by fluorescence in situ hybridization (FISH), inductively coupled plasma-optical emission spectroscopy (ICP-OES), Fourier transform-infrared spectroscopy (FTIR), fluorescein iso-thiocyanate (FITC) labeling, etc. Subsequently, the inherent mechanisms of EPSs that affect metal oxide NPs’ fates and their interactions that potentially affect the biodiversity and structure of complex microbial communities are discussed. This study provides a better understanding of the toxicity mechanisms of different metal oxide NPs on AGS, the roles of EPSs in protecting microbial cells against NPs exposure, and illustrates the potential fate and risks of NPs during biological wastewater treatment.

## 2. Materials and Methods

### 2.1. Inoculum and Substrate

The methanogenic seed AGS used in this study was withdrawn from a full-scale expanded granular sludge bed (EGSB) anaerobic reactor treating soybean wastewater (Harbin, China). The ratio of volatile suspended solids (VSSs) to total suspended solids (TSSs) was 87.30%. Initially, the seed AGS was acclimated to synthetic wastewater (Appendix A) under 35 ± 1 °C for approximately one month until biogas production and methane content reached a steady state. After the acclimation phase, assays utilizing acetate and hydrogen as the substrate were conducted that revealed the maximum methanogenic activity was 587.3 ± 39.6 and 947.93 ± 85.9 mgCH_4_-COD/(gVSS·d), respectively. The natural concentrations of titanium, zinc, and copper in the AGS were 1.7 ± 0.3, 4.1 ± 0.8, and 0.6 ± 0.1 mg/gVSS, respectively.

### 2.2. Metal Oxide NPs and Their Dissolved Metal Ions

Three commercial metal oxide NPs (TiO_2_, ZnO, and CuO) were purchased from Sigma-Aldrich (St. Louis, MO, USA) as dry powders with an average size of less than 50 nm and purity higher than 99%.

Stock suspensions (10 g/L) were prepared on the day of the experiment by adding 10 g NPs to 1 L of ultrapure water containing polyacrylamide, which is one of the most common and low-cytotoxicity dispersants [16], to avoid NP aggregation, followed by sonicating for 2 h (KUDOS SK3300H, 180 W, 53 kHz, Shanghai, China). Stocks were stored at 4 °C and sonicated for 1 h before use. Analysis of the suspension by dynamic light scattering (DLS) using a Zetasizer Nano ZS (Malvern Inc., Westborough, MA, USA) indicated that the average particle sizes of TiO_2_, ZnO, and CuO NPs were approximately 136 ± 42 nm, 175 ± 47 nm, and 112 ± 38 nm, respectively.

To determine the soluble fraction of NPs (Zn^2+^ and Cu^2+^), NPs stock suspensions were diluted (as needed) with synthetic wastewater as described in Section 2.1 to reach the desired initial concentrations (10, 20, 50, 100, 150, and 200 mg/L). All of the above NP solutions were maintained in an incubator shaker at 110 rpm and 35 °C for 48 h. At different intervals, mixtures samples were withdrawn synchronously and centrifuged at 12,000 rpm for 10 min. The resulting supernatants were filtered through 25 nm mixed cellulose ester membrane and acidified with 4% ultrahigh purity HNO_3_ prior to inductively coupled plasma-optical emission spectroscopy (ICP-OES; Optima 2100 DV, Perkin Elmer, Waltham, MA, USA) analyses.

### 2.3. Methanogenic Activity Bioassays and Testing of NPs Toxicity to AGS

All bioassays were carried out under mesophilic (35 ± 1 °C) conditions in a dark room using a series of serum bottles (150 mL). The 1.08-times concentrated medium (37 mL), which was the same as that in the parent anaerobic reactor without glucose, was added to each bottle under a gas mixture N_2_/CO_2_ (80:20, *v*/*v*) atmosphere. After sieving through 70-mesh sieve to remove fine particles, the AGS was withdrawn from the parent reactor and inoculated in the serum bottles to a final concentration of 1000 mgVSS/L. Either sodium acetate (1000 mgCOD/L) or H_2_ gas were used as a substrate. H_2_ was supplied at an initial concentration of 0.5 atm of H_2_/CO_2_ (80:20, *v*/*v*) overpressure. Subsequently, all assays were pre-incubated overnight to avoid a lag phase caused by the atmospheric oxygen exposure.

After pre-incubation, 3 mL of appropriately diluted NP stocks was spiked into bottles to obtain a series of final mass concentrations ranging from 10 to 200 mg/L, and 3 mL of deionized water without any chemical was added to the control bottles. A total mixed liquor volume of 40 mL was kept in each bottle. Then, all bottles were flushed with N_2_/CO_2_ (80:20, *v*/*v*) gas mixture, and H_2_/CO_2_ (80:20, *v*/*v*) was added again to the hydrogenotrophic assays to a final pressure of 0.5 atmH_2_. Both pre-incubation and incubation were performed in an incubator shaker at 110 rpm.

The production of headspace CH_4_ biogas was measured at given time intervals until the theoretical maximum methane production was reached. Then, the maximum specific methanogenic activities (SMA, gCH_4_-COD/(gVSS·d)) were calculated from the slope of the CH_4_ content versus time graph. The normalized methanogenic activity (NMA) in the assays exposed to different NPs was calculated as the percentage of the ratio of SMA of the control without NPs as shown below:(1)$$\mathrm{NMA}\;\left(\%\right)=\frac{{\mathrm{Maximum}\;\mathrm{SMA}}_{\mathrm{NPs}}}{{\mathrm{Maximum}\;\mathrm{SMA}}_{\mathrm{control}}}\times 100\%$$

The initial concentrations of NPs causing 50% inhibition (IC50) were calculated by interpolation or extrapolation of graphs plotting the NMA as a function of the NPs’ concentration.

The experiments on the effect of the released ions from NPs on methanogens were conducted using the same conditions as described above, except that the corresponding dissolved metal ions determined in Section 2.2 were used to replace the NPs.

### 2.4. EPS Extraction and Analysis

Considering the environmentally relevant concentration of NPs and their high bioaccumulation in activated sludge, three representative concentrations (10, 50, and 200 mg/gVSS) were chosen to investigate the role of EPSs in the response of AGS to the toxicity of different NPs; the experimental conditions are described in Section 2.3. In this case, glucose was used as the carbon source because of its easy degradability to hydrogen and acetate, which were then both used as substrates for acetoclastic and hydrogenotrophic methanogens.

At the end of the batch experiment, a modified heat extraction method was applied to extract the loosely bound EPS (LB-EPS) and the tightly bound EPS (TB-EPS) from the AGS withdrawn from the NP-containing and control serum bottles, respectively. A detailed description of the EPS extraction process is provided in Appendix A. Using colorimetric methods performed with a UV/vis Spectral Photometer (INESA, Shanghai, China) according to the reference, both the extracted LB-EPS and TB-EPS fractions were analyzed to determine the content of proteins, polysaccharides, and DNA.

The distribution of NPs in the reaction system was investigated by measuring the Ti, Zn, and Cu contents in the liquid and EPS extractions by ICP-OES following acid digestion according to the procedures documented in the previous literature [17]. The immobilized NPs by AGS were calculated as the difference between the added quantity of NPs and the sum of the measured NPs’ quantity in the supernatant and EPS extraction

The extracted EPS solution was freeze-dried for 48 h and subsequently analyzed using a Fourier transform-infrared spectroscopy (FTIR) spectrophotometer (Spectrum One, Perkin-Elmer, USA) for determination of the alterations of the major functional groups of EPS produced due to the presence of different NPs.

### 2.5. Microbial Community Analysis

The distribution of the main groups of microorganisms in AGS after NP exposure (Section 2.3) was analyzed by the fluorescence in situ hybridization (FISH) technique using an oligonucleotide probe EUB338-mix and ARC915, to identify presumptively acidogens/acetogens and methanogens, respectively [18]. The probes were 5′ end labeled with Cy3 and fluorescein iso-thiocyanate (FITC), respectively; the detailed procedure of the FISH analysis is presented in the Supplementary Information (Appendix A). After the FISH procedure, some samples were stained with DAPI to quantify the relative proportion of the bacterial (EUB338) and archaeal (ARC915) cells by the direct counting method [19], and the results were analyzed with an image analysis system (Image-Pro Plus, V6.0).

### 2.6. Other Analytical Methods and Statistical Analysis

The production of reactive oxygen species (ROS) and the level of extracellular lactate dehydrogenase (LDH) were measured after the exposure experiments, described in Section 2.3, to investigate the possible toxic mechanism of different NPs; the procedure is detailed in the Supplementary Information.

TiO_2_, ZnO, and CuO NPs were labeled with FITC for visualization of NPs’ distribution in AGS, respectively, following the protocol of Bellio et al. [20]. Details of the procedures are described in the Supplementary Information (Appendix A). After labeling, the serum bottles, which contained AGS and medium as described in Section 2.4, were added with weighed amounts of TiO_2_, ZnO, and CuO NPs as dry powder to a final concentration of 200 mg/gVSS, respectively. Subsequently, the adsorption-equilibrated AGS samples were collected and observed under a fluorescence microscope (BX51, OLYMPUS, Tokyo, Japan) at an excitation/emission wavelength of 488 nm/525 nm.

One-way analysis of variance was used to determine the statistical significance of the differences between values. All the above tests were performed in triplicate and the results are expressed as (mean values ± standard error) in both figures and tables.Univariate logistic regression analysis was conducted to test the association between the different variations, and *p* < 0.05 and a correlation coefficient of more than 0.8 (positive or negative) were considered to indicate a statistically significant correlation; asterisks are used to indicate statistical differences (*p* < 0.05) from the control in the figures.

## 3. Results and Discussion

### 3.1. Effects of NPs Exposure on Methanogenic Activity

In this study, 10 mg/L polyacrylamide was used as a dispersant. To exclude incidental biological effects of dispersant on AGS, samples with or without polyacrylamide were firstly tested, and the results are stated in Appendix A. According to the EPS and methane production results, 10 mg/L polyacrylamide does not exert toxic effects on microorganisms in AGS, but helps to form homogeneous dispersions, if taking the stable suspension of NPs measured by DLS (Section 2.2) into consideration. Thus, hereafter, the term “control assay” refers to the addition of polyacrylamide rather than NPs.

Methanogenesis is often considered to be the most sensitive step in the anaerobic digestion process [21]. Two different assay substrates were tested separately to assess the effects of NPs and their corresponding released ions on the activity of acetoclastic and hydrogenotrophic methanogens (Figure 1). Since the low solubility of TiO_2_ NPs has been the subject of a number of reports and publications, the toxicity of Ti^4+^ on methanogens was not investigated.

The NMA values under either acetoclastic or hydrogenotrophic methanogenic conditions were above 90% and did not show evidence of an increasing toxicity effect over TiO_2_ NP dosages. These results indicated that the TiO_2_ NPs cannot be considered toxic to methanogenesis at the concentrations studied, which is consistent with the reports of other research groups [22]. The negative influence of both ZnO and CuO NPs on the methane production rate was dose dependent. However, CuO NPs showed higher toxicity to methanogens than ZnO NPs. Thus, initial concentrations of CuO NPs used in the assays that were lower than those of ZnO NPs caused the same or even greater toxic effects on the methanogenic activity. The hydrogenotrophic and acetoclastic NMA values decreased by 23.1% and 45.3% at ZnO NPs dosages of 200 mg/L, whereas the corresponding parts by CuO NPs decreased by 49.4% and 77.0% respectively at the above tested dosages. Previous investigations also indicated that CuO NPs exerted more pronounced toxic effects on methanogenesis than ZnO NPs [23]. In contrast to the results in this research, other studies found greater sensitivity of methanogens to the presence of ZnO NPs as compared to CuO NPs [24]. Since different types of anaerobic sludge (flocculent or granular sludge) seemed to be used in the above papers, these discrepancies indicated that the planktonic form may be an important factor in the response of AGS to NPs’ toxicity [14].

Figure 1 also provided information on the susceptibility of acetoclastic and hydrogenotrophic methanogens to metal oxide NPs’ exposure. Under all the NPs tested and incubated, the hydrogenotrophic methanogens were consistently less susceptible to the inhibition of NPs than acetoclastic methanogens. The IC50 values for the acetoclastic assays were estimated to be 214 mg/L of ZnO and for 66 mg/L of CuO NPs, whereas in the hydrogenotrophic assays, the IC50 values of ZnO and CuO NPs were above 1000 and 179 mg/L, respectively. Thus, it can be concluded that acetoclastic methanogens were inhibited by NPs to a larger extent than hydrogenotrophic methanogens, this is in agreement with previous findings and has important implications since about 70% of methane formed in an anaerobic digester is derived from acetate [21].

Several studies concluded that the toxicity of ZnO or CuO NPs to anaerobic digestion microorganisms can be attributed to the higher solubility and thus the easier release of Zn^2+^ or Cu^2+^ in the test medium [25]. Thus, the dissolved Cu and Zn concentration and the impacts of the released ions on NMA were also determined by the batch toxicity assays. It was found that the solubility of ZnO NPs was much higher than that of CuO NPs, which led to different effects on NMA. At any given soluble concentration of Zn, a similar effect on the NMAs was observed, irrespective of whether the ions were released from ZnO NPs or remained from ZnCl_2_ salt used, which may suggest that the Zn^2+^ release was the main mechanism of ZnO NPs’ toxicity on methanogens. However, the released Cu^2+^ concentrations, even under a high dosage of 200 mg/L CuO NPs, had inhibitory effects that were far less pronounced than those of CuO NPs (Figure 1c). It seems that the toxicity of CuO NPs can only be partially explained by the dissolution of CuO NPs to Cu^2+^. Previous results were also unable to explain the toxicity by dissolution of CuO NPs [25].

### 3.2. Effect of NPs Exposure on EPS Production

As an important component of AGS for the matrix structure build-up and the buffer towards an external harsh environment, EPS is often fractionated to LB-EPS and TB-EPS [26]. Since the magnified skin layer of a granule shows that EPS was within the entire granular structure, and numerous bacterial rods, cocci, and filaments of various sizes were surrounded, EPS was believed to be the main integral part that contacted with NPs (Appendix A) and its production depended considerably on the different environmental challenges [27]; thus, the effects of the NP type and the selected dosage on EPS secretion were investigated (Figure 2).

In general, the production of TB-EPS was greater than that of LB-EPS, regardless of the NPs’ type and dosage. However, the content of both LB-EPS and TB-EPS appeared to be strongly affected by the exposure to different NPs. The average content of LB-EPS, TB-EPS, and total EPS increased by 69.5%, 38.8%, and 48.4% (calculated based on the test results), respectively, as the concentration of TiO_2_ NPs increased from 0 to 200 mg/L, indicating that TiO_2_ NPs can stimulate EPS generation. By comparison, the low concentrations of ZnO NPs (10 mg/L) evidenced their negative impact on LB-EPS content. The increase in the ZnO NPs’ dosage to 50 mg/L, however, resulted in a significant rise in the LB-EPS level, which started to decline as the dosage was elevated to 200 mg/L. Inhibitory responses concerning TB-EPS were manifested only at ZnO concentrations of 200 mg/L, whereas in the CuO NPs assays, significant inhibition was observed at all tested dosages except for that of LB-EPS under 10 mg/L CuO NP exposure, which slightly increased from 34.8 mg/gVSS (control) to 38.0 mg/gVSS.

The above results suggest that the production of EPS by AGS was highly dependent on the NPs type and dosage, which can be explained by the potential protective role of LB-EPS and TB-EPS, in addition to the different toxicity mechanism of NPs to AGS. Since low bacterial cytotoxicity of TiO_2_ NPs under a dark environment due to its poor solubility was confirmed by Figure 1 and other findings [28], the increase in EPS upon exposure to TiO_2_ NPs may be attributed to the AGS self-protection from the direct effect caused by physical restraints [29]. As a primary surface for contact with the NPs, LB-EPS can provide a large number of binding sites to trap NPs outside the cell membrane [8]. Consequently, AGS has to generate more LB-EPS to compensate for the occupied binding sites to maintain proper mass transport. The series of SEM images in Appendix A also shows the presence of a large number of nodules on the AGS surface, indicating the biosorption of NPs by AGS, and the EDAX analysis conclusively identified them as Cu, Zn, and Ti. The FITC staining profiles in Appendix A illustrated that NPs are mainly located superficially in the AGS structure, which is consistent with the previous studies [30].

In contrast, TB-EPS tended to accumulate and form tightly packed granular structures to either delay or prevent TiO_2_ NPs from reaching the inner microbes [31]. The increase in EPS content under low-cytotoxic NP exposure was also reported by other authors [32]. At a certain threshold, Zn^2+^ can increase methanogenic activity and thus higher organic loading rates are obligatory [33]. Previous studies also confirmed that a high level of EPS is not beneficial for substrate mass transfer [34]. Therefore, the slightly lower EPS levels at the dosage of 10 mg/L ZnO NPs may be due to a self-adjustment to accelerate the import of nutrients and the export of metabolic products [13]. Although the elevated EPS production seemed to be an effective strategy for mitigation of the toxic effect of 50 mg/L of ZnO NPs, higher ZnO NP dosages (200 mg/L) may have exceeded the shielding capacity of EPS. Thus, the metabolism of microorganisms in AGS became repressed, which led to a decrease in the corresponding EPS generation. In contrast, the decreased EPS content caused by all tested dosages of CuO NPs suggests that the microorganisms in AGS can be negatively influenced by a low level of CuO NPs and the protective capability of EPS became less significant. The remarkable discrepancy in EPS content under different NP exposures provides further evidence that there are significant essential differences between the toxicity mechanisms of metal NPs to AGS, which vary depending on the NPs’ type and dosage.

According to the aforementioned data, NPs’ effects on the methanogenesis and EPS content can be associated with their fate and space distributions in AGS. In view of this, the NPs’ distribution in different phases was investigated, as shown in Table 1.

LB-EPS can absorb more NPs than TB-EPS, indicating that the biosorption affinity of TB-EPS to NPs is less important than that of LB-EPS. The quantity of TiO_2_ and ZnO NPs immobilized by EPS was higher than that in wastewater solution or AGS, indicating the effective protective capability of EPS against TiO_2_ and ZnO NPs. In contrast, however, more CuO NPs ended in AGS, and the distribution ratio of copper immobilized by AGS was increased with the increase in the CuO NP dosages, whereas it decreased in EPS, indicating EPS cannot provide enough protection for AGS. Thus, the microbes’ cells can be directly exposed to high toxic levels of CuO NPs, which may lead to the observed repression of EPS secretion as described in Figure 2.

### 3.3. Possible Effect of NPs on Cytomembrane Integrity

ROS production and LDH release induced by metal oxide NPs are related to the depression of bacterial cell viability, and, therefore, in this study, these two parameters were measured to explore the potential toxic mechanism of CuO NPs towards AGS. As shown in Figure 3, a significant increase in the intracellular ROS production was observed only in the presence of a high dosage of CuO NPs, which indicated that the antibacterial activity of CuO NPs may be partly related to ROS production. From the results showing that the EPS content was significantly inhibited by the CuO NPs, but not by TiO_2_ and ZnO NPs (Figure 2), it can be suggested that EPS acted as a protective barrier by keeping TiO_2_ and ZnO NPs distant from the cells; however, this effect was not applicable to CuO NPs. This also indicated that there must be a different toxic mechanism between CuO NPs and the other two NPs.

Although some researchers speculated that the generation of ROS by NPs was limited under anaerobic conditions [35], the findings of the present and other studies have shown that ROS can be detected in the absence of oxygen or light [36]. Considering the inadequate secretion of EPS, microorganisms in AGS may directly interact with CuO NPs and thus be negatively affected by ROS. However, these levels of ROS production were not sufficiently high to cause the strong inhibition illustrated in Figure 1c, which indicates that there were other factors besides ROS that contribute to the CuO NPs’ toxicity. Combined with the results showing the significant increase in the EPS production in the treatments with TiO_2_ NPs (Figure 2), it can be hypothesized that the low intracellular ROS production induced by TiO_2_ NPs cannot damage the EPS, or even the microorganisms cells embedded in it, as it is well known that ROS have exceedingly short lifetimes of approximately 3 ns and are only effective if produced close to or inside the cell membrane [37]. The difference in the increase in ROS generated by ZnO NPs was not statistically significant from that in the control treatment (*p* > 0.05), indicating that ROS were not mainly responsible for the cytotoxicity of ZnO NPs.

LDH release is an indicator of the cell membrane integrity. A significant increase in LDH release was observed only in the treatment with the addition of CuO NPs, indicating that cell leakage occurred. Since the contribution of ROS production to membrane damage can be partly excluded according to the above discussion, and the damage of the cell membrane was not caused by the Cu^2+^ release (Figure 1c), it can be concluded that the most possible mechanism may have been the direct physical interaction between the microorganism cell in AGS and CuO NPs, which severely compromised the integrity of the cytoplasmic membrane.

It was also found that TiO_2_ and ZnO NPs did not cause a considerable increase in LDH, and ZnO NPs even led to LDH reduction at the dosage of 10 mg/gVSS, which suggests that these NPs do not exert toxicity on the cell membrane structure. Zn^2+^ is one of the essential metals for the biochemical process in all organisms since it is crucial for the structure of cell membranes and key cellular processes. Thus, the release of a proper quantity of Zn^2+^ from ZnO NPs, which may benefit the metabolism of microorganisms in AGS, may explain the temporary reduction in LDH. Considering that no significant membrane damage occurred under ZnO exposure, even though direct physical interaction also existed between the AGS cell and ZnO NPs according to Table 1, it can be speculated that the inhibition of ZnO NPs on AGS mainly occurred inside the cells. Since former studies found that NPs with a size larger than 10 nm would be excluded from biofilms, and Zn^2+^ could be readily taken up by membrane transporters, the former conclusion that the release of Zn^2+^ was the main reason for the inhibitory effects of ZnO NPs on the methane production was further verified.

### 3.4. The Role of EPS on the Diversity and Abundance of Microorganisms in AGS under NPs Exposure

The above results indicate that the NPs can impose distinctive adverse effects on AGS and the promotion of EPS production can alleviate the toxic effects of certain NPs. However, to date, the strategies that provide the unique resistance by EPS to the toxic influence of different metal oxide NPs are still unclear. It is reported that metal oxide NPs have a negative effect on the microbial diversity and composition of the bacterial community [38]. Moreover, other researchers have discovered that microbial cells in activated sludge systems can alter the structure and composition of EPS to improve their resistance to NPs [37]. Thus, in this study, the role of EPS on the diversity and abundance of sludge microorganisms in AGS under NP exposure was preliminary explored by determining the changes in the EPS composition and the microbial community via FISH analysis, and the results are shown in Figure 4. 

Protein was the major component of the EPS matrix, which was also subjected to the most pronounced change amongst the compounds of the EPS biopolymer fraction with the presence of metal oxide NPs; in contrast, no significant differences in the polysaccharides content were observed among all tested NPs, indicating that the role of proteins in EPS was relatively more important to the alleviation of NPs’ toxicity than that of polysaccharides. Previous studies have revealed that, besides offering a large quantity of exoenzymes for the hydrolysis and transport of the organic matter, the protein in EPS also induces aggregation of NPs and efficiently localizes NPs away from cells [4]. Joshi, Ngwenya and French [37] found that no change in the size of aggregates was evident with or without the addition of polysaccharides to Ag NPs aqueous dispersion. Combining the findings confirmed by SEM images and NP distribution assays in this study, it can be rationally speculated that the secretion levels of EPS protein were closely linked with the response of the microbial community to NP exposure, and this hypothesis was confirmed by correlation analysis between EPS composition and abundance of bacteria and archaea, as shown in Appendix A.

It was found that the abundance of methanogenic archaea was significantly increased by high dosages of TiO_2_ NPs, whereas the abundance of acidogens, which was detected by the EUB338 probe, decreased with the increase in TiO_2_ NP content. Obviously, TiO_2_ NPs had a stronger negative effect on acidogens rather than on methanogens. A possible explanation may be associated with the multi-layered structure of AGS described earlier; that is, since TiO_2_ NPs’ aggregates were unlikely to have diffused to the internal granular layer due to the protection of the EPS matrix, only acidogens located in the AGS surface had the opportunity to contact with TiO_2_ NPs. Thus, their metabolism may have been restricted because of the occupancy or blockage of the substrate transfer path by TiO_2_ NPs. The increase in the protein content can be regarded as a resistance mechanism of acidogens against the limitation of nutrient availability, which may explain why the close relations between microbial abundance and EPS composition were only observed in acidogens and EPS protein (see Appendix A). In the core of the granule, a shelter for methanogens from TiO_2_ NPs exposure was provided by both EPS and acidogens. Furthermore, metabolic depression of acidogens caused by TiO_2_ NPs may have led to a lower volatile fatty acids (VFAs) loading, which may have been beneficial for the slow-growing methanogenic archaea. This phenomenon may provide some explanations for the insensitivity of methanogens to TiO_2_ NPs, thus resulting in a significant increase in their abundance.

According to previous researchers, the promotion of EPS production has been usually observed under environmental stress. Therefore, the decrease in the content of proteins and polysaccharides, in addition to the methanogenic archaea dominance, indicated the formation of favorable conditions for methanogenesis in AGS by the addition of 10 mg/L ZnO NPs, which was consistent with the aforementioned results. However, completely opposite changes in the microbial community and EPS content occurred when the ZnO NPs dosage was increased to 50 mg/L, suggesting that methanogenic archaea were more sensitive to ZnO NPs than acidogens and the protection of EPS was inadequate. In the treatments with 200 mg/L ZnO NPs, the gap between the abundance of acidogens and that of methanogenic archaea was increasingly widened, and the protein production was further decreased while polysaccharides approached the control level, indicating that the limited protective capability of EPS can be attributed to instability or sensitivity of the EPS protein. One of the key functions of EPS protein is its involvement in the enzymes performing the digestion of macromolecules and particulate material in the microenvironment of the embedded cells. By comparison, previous studies have proven that the antimicrobial toxicity of Zn^2+^ can be due to the blockage of intracellular enzyme functions, especially in the methanogenic cell [39]. The EPSs normally contain small quantities of DNA, and large quantities of DNA in the EPS can be an alarming indication of cell lysis and the release of intracellular DNA. Analytical results show that each gram of VSS contained 0.10–6.18 mg of DNA, which accounted for only 1~6% of the EPS extracted after ZnO NP exposure (Figure 4b). All the evidence suggests that the effect of ZnO NPs on AGS can be attributed to Zn^2+^, which was transferred to intracellular space by the protein of EPS.

CuO NPs also induced a significant decrease in EPS protein concentration and archaea abundance. The DNA content, however, was much higher than that under ZnO NP exposure, implying that CuO NPs caused cell breakage and release of intracellular contents. Combined with the small amount of copper ion release and the ROS production, which was insufficient to trigger the severe inhibition by CuO NPs, the possibility that the toxic effects occurred inside or outside of the cell membrane could be excluded. Thus, the inhibition of CuO NPs on AGS microorganisms seemed to be manifested more obviously on the cell membrane, which can be explained as follows: the CuO NPs absorbed by AGS initially inhibited the synthesis of EPS protein, which led to destabilization of the EPS structure, and thus EPS could not completely impede the access of CuO NPs to the intra-granular space. CuO NPs not only directly contacted acidogenic biofilm on the outer surface of AGS, but also diffused to deeper positions in AGS through the cracks or micro-channels in EPS used for the diffusion of substrates and metabolites into and out of the granules, where the methanogenic biofilm was located. Hence, the cell membrane of both acidogens and methanogens may interact with CuO NPs, and consequently, membrane perturbation was induced. Similarly, Fu et al. [40] stated that Ag NPs can penetrate into the structure of the aerobic granular sludge and inhibit the activity of the denitrifying bacteria that were located inside the granules.

The above results indicate that EPS exhibited strong affinities towards NPs, which are one of determinants for NPs’ mobility and toxicity; thus, FTIR analysis was performed to reveal the functional groups of sludge EPS and possible participation in biosorption. The FTIR spectrum of the EPS is shown in Appendix A, and the main functional groups observed with the FTIR spectra are summarized in Appendix A.

The FTIR spectra of the EPS with the presence of ZnO or CuO NPs were similar. However, after exposure to the TiO_2_ NPs, the spectrum showed some different functional characteristics, and all these spectra demonstrated some qualitative differences from that of the control, suggesting that different functional groups of EPS fraction were affected by the addition of NPs, which agreed well with the results of other authors [41]. Among all the functional groups, those involved in protein had the closest relations with the NPs’ exposure with an apparent shift or weakness in peaks. The intensity of the peak at 1655 cm^−1^, which was assigned to the amide-I bands of proteins, was reduced, whereas the bands at 1581 cm^−1^, which also originated from peptidic bonds, arose after exposure to the ZnO or CuO NPs. The peak at 1462 cm^−1^ in the control sample, representing the bending vibrations of C-H and the stretching vibrations of C-N, which were considered to be the amide III of a protein-like substance, became weak and shifted to 1428 cm^−1^ after ZnO or CuO NP exposure. This may imply the structure variation in the protein in EPS after absorbing ZnO or CuO NPs. The appearance of the weak band at 1415 cm^−1^ was only observed in the EPS spectrum in the TiO_2_ NP sample, which may be due to the excessive productivity of EPS protein after TiO_2_ NP adsorption. The band at 1125 cm^−1^, assigned to C−O−C of polysaccharides, showed a shift to 1106 cm^−1^ with little intensity changes, which was in agreement with the above results that showed that the adsorption of NPs was not mediated by polysaccharides. Some bands at the “fingerprint zone” (881 cm^−1^) were assigned to the phosphate group, which is one of the functional groups from which nucleic acids are composed [42]. This phenomenon showed agreement with the increase in DNA in EPS after CuO NPs’ exposure.

However, it should be noted that this work just confirmed that EPS secreted by AGS can affect the fate of metal oxide NPs, and their interactions result in huge differences in the biodiversity and structure of complex anaerobic microbial communities. Comprehensive investigations of the various species of cells comprising the biomass, the impacts of microbial EPS on the chemical transformations of NPs, the hetero-aggregation characteristics of NPs with AGS, and the formation and element compositions of the EPS–NP complex are still not clear, and thus require in-depth investigation in future study.

## 4. Conclusions

Metal oxide NPs are commonly and extensively used; therefore, they are likely to be found in WWTPs. Nonetheless, their effect on anaerobic biological treatment microbial diversity has been poorly described. In this research, AGS was applied to systematically examine the potential fate and risks of NPs in anaerobic reactors. The research emphasis was on the metabolic capabilities, the physiological profiles of the microbial community, and the importance of EPS as a microbial protective barrier in NP-affected anaerobic treatment systems.

The results of this study showed that:(1)The sensitivity of methanogens to each type of metal oxide NP varied with the species. CuO NPs were the most toxic among the studied NPs, since methanogenic activity showed a significant inhibitory effect, consistent over the test period, with the IC50 of 65.7–179 mg/L for CuO NPs and 214 to over 1000 mg/L for ZnO NPs.(2)The average content of LB-EPS, TB-EPS, and total EPS increased by 69.5%, 38.8%, and 48.4% (calculated based on the test results), respectively, as the concentration of TiO_2_ NPs increased to 200 mg/L, which provided evidence that AGS prevented the physical restraints caused by TiO_2_ NPs through secreting more EPS. There is a disparity between the variation trend of LB-EPS and TB-EPS at a ZnO NP exposure concentration of <200 mg/L, which may be explained by the Zn^2+^ release.(3)The presence of CuO NPs promoted the ROS generation and LDH leakage of AGS. Thus, the negative impact of NPs was caused not only by the dissolved metal ions, but also by their considerable potential to induce physical restraints or membrane reduction in AGS cells without the presence of sufficient protective effects exerted by EPS.

The present results can offer useful information to understand the effect of metal oxide NPs on the performance and microbial community of anaerobic wastewater biological treatment systems, and help researchers recognize the important contributions of EPS. This which could provide new insights for evaluating and migrating the potential risks of NPs in WWTPs.

## Figures and Tables

**Figure 1 ijerph-19-05371-f001:**
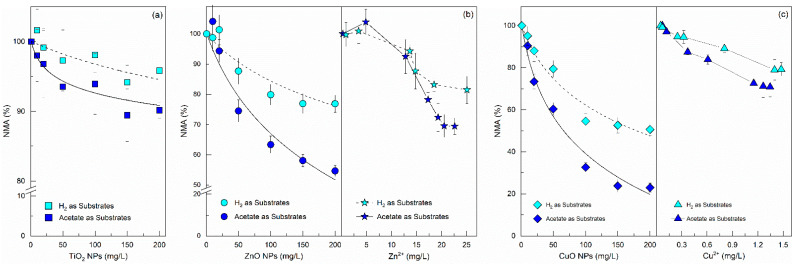
NMA determined in batch toxicity assays as a function of the initial concentration of TiO_2_ NPs (**a**), ZnO NPs (**b**), or CuO NPs (**c**) using acetate or hydrogen as substrates; NMA is presented as a function of the final concentration of their corresponding dissolved Zn^2+^ or Cu^2+^, which was calculated from Equation (1). Error bars represent standard deviations of triplicate tests.

**Figure 2 ijerph-19-05371-f002:**
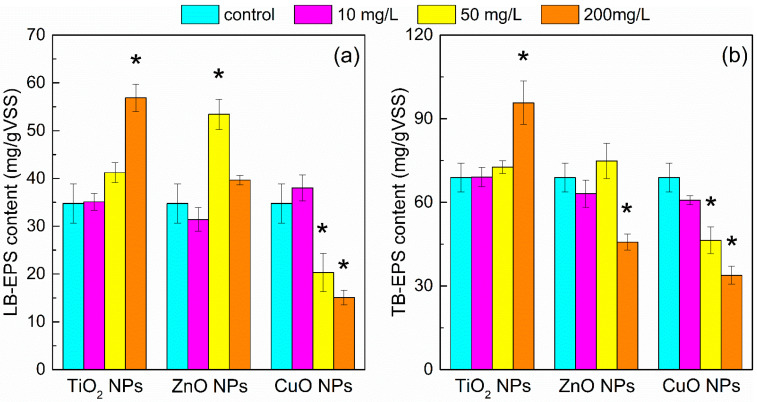
Effects of NPs’ exposure on the production of LB-EPS (**a**) and TB-EPS (**b**). Asterisks indicate statistical differences (*p* < 0.05) from the control. * Error bars represent standard deviations determined from triplicate measurements.

**Figure 3 ijerph-19-05371-f003:**
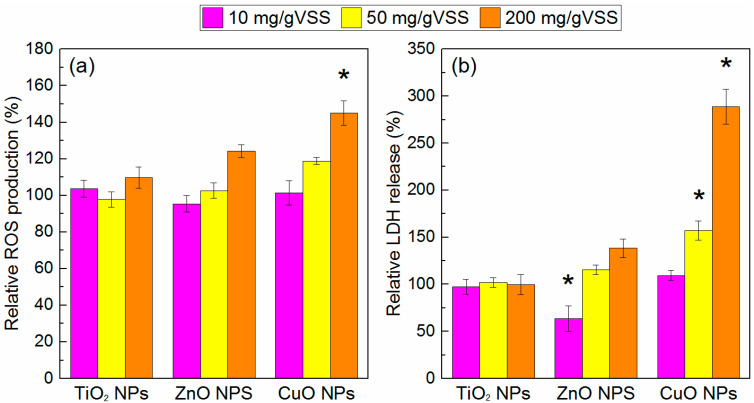
Relative ROS production and LDH release in AGS exposed to different concentrations of NPs. Asterisks indicate statistically significant differences (*p* < 0.05) from the control. * Error bars represent standard deviations of triplicate measurements.

**Figure 4 ijerph-19-05371-f004:**
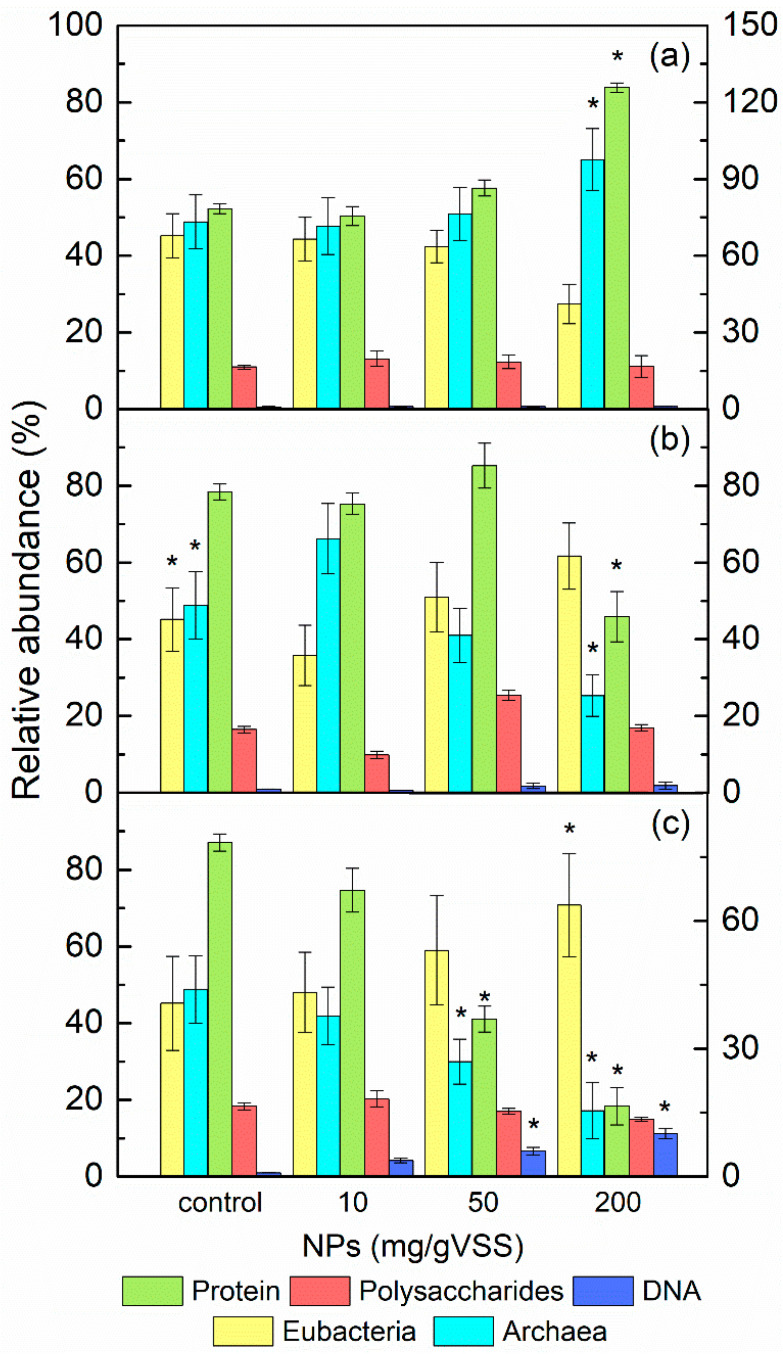
Comparisons of the microbial community structure and EPS compositions in AGS after TiO_2_ (**a**), ZnO (**b**), and CuO (**c**) NPs’ exposure. * Error bars of the relative abundance and EPS content represent standard deviations of the triplicate slot blot hybridizations and triplicate measurements, respectively. Asterisks indicate statistically significant differences (*p* < 0.05) from the control.

**Table 1 ijerph-19-05371-t001:** The fate of different dosages of NPs in supernatant and AGS ^a^.

NPs		Supernatant	EPS		Immobilized AGS
Type	Concentration		LB	TB	
TiO_2_	10	0.73 ± 0.05	7.34 ± 0.25	0.62 ± 0.11	1.32 ± 0.06
	50	3.35 ± 0.15	28.53 ± 1.35	5.29 ± 0.24	12.84 ± 0.58
	200	16.60 ± 0.69	85.66 ± 3.66	31.06±1.32	66.68 ± 3.13
ZnO	10	1.14 ± 0.04	6.22 ± 0.28	1.18 ± 0.45	1.46 ± 0.06
	50	6.35 ± 0.29	21.88 ± 0.97	11.51 ± 0.52	10.27 ± 0.49
	200	16.60 ± 0.72	67.69 ± 3.28	32.83 ± 1.56	82.88 ± 3.72
CuO	10	0.37 ± 0.02	7.12 ± 0.33	0.95 ± 0.05	1.56 ± 0.06
	50	1.36 ± 0.06	16.74 ± 0.79	9.28 ± 0.46	22.64 ± 1.12
	200	10.08 ± 0.44	51.83 ± 2.33	20.12 ± 0.97	117.98 ± 5.57

^a^ The data reported are the averages and their standard deviations determined from triplicate measurements, and the unit is mg/gVSS.

## Data Availability

The data that support the findings of this study are available from the corresponding author.

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
