# Peer review of "The Role of Extracellular Polymeric Substances in the Toxicity Response of Anaerobic Granule Sludge to Different Metal Oxide Nanoparticles"

_ijerph, 2022, doi:10.3390/ijerph19095371_

Round 1
Reviewer 1 Report
Abstract is not mentioning in brief how the experiments were conducted. Just one or two sentences briefly explaining how the investigations were carried out will be sufficient
Line 67: ….... “Regarding to this point by now” should be…regarding this point by now
Line 114 to 116: Three commercial metal oxides NPs (TiO2, ZnO, and CuO) were purchased from 114 Sigma-Aldrich (St. Louis, MO, USA) as dry powders with an average size of less than 50 115 nm and purity higher than 99%> . Can authors clarify if the metal oxides were analytical grade or further purification was performed first before usage
Line 116: The average particle size of and surface morphology CuO should be…The average particle size and surface morphology of CuO
Line 130:… withdrawn and centrifuged at 12,000 g for 10 min…Its assumed the 12000 g is supposed to be 12000 rpm
Line 136: ….The 1.08× concentrated medium…the x, is there something missing or it denotes “1.08 times concentrated medium…if this is the case then better use words (times) instead of the sign (x)
Line 180: ….li terature should be literature
222 to 226: Figure 1, the name is too long and seems to be having different font sizes.
Line 248: In Contrast to our results,…would advise avoiding personalizing the writing…rather, In contrast to results in this paper/research..
Line 249….typea….type
Line 279 to 281: Figure 2 the name is too long. The description of error bars could be avoided as consumers of scientific literature should already be knowing the meaning error bars
Line 305: Rather remove “our” so the sentence becomes…..been confirmed by Figure 1 and….
Line 329 to 331: The Figure 3, the asterisks explanation could be included in the text rather than under the figure as it makes the name to be too long
Line 382 to 385: Figure, the name is too long. The description of error bars could be avoided as consumers of scientific literature should already be knowing the meaning error bars. The asterisks part could be included in the text rather than under the figure as it makes the name too long
Author Response
There is a quantity limitation for the upload files, so we had to merge the Reply, the revised manuscript and the revised Supplementary Information into one PDF file, please check it in the attachment below.

Reviewer 2 Report
Please find below the comments which could improve the quality of the paper.
- Line 16 abstract TiO2 not TiO2
- Try to avoid abbreviations like EPS, AGS, ROS, LDH ect in the abstract
- Line 21 apt what does that mean
- Line 32 “As a result, Concerns” why concerns start with capital letter
- Line 125 what are the measurement units of NP “136 ± 42, 175 ± 47 and 112 ± 38”
- Check line 198 is it ROS first time used in the text. Please provide in full words when first-time use and abbreviation into brackets later in the text use the abbreviation
- Figure 1 the figures don’t look correctly when 200 value is on the frame or 0 value of the other figure. Please improve figure 1.
- Line 249 “typea”
- Line 305 what do you mean other findings. Please be more specific
- What was the control treatment when talking about ROS measurements
- Figure2 GVSS in the key of the figure what does that mean or it is a mistake?
- In figure 3 (b) the huge amount of LDH release in the treatment with Cu NPs means that Cu NPs show low toxicity for LDH how you could interpret these results.
- Figure 4 the key of the figure too small, please increase the font. It is needed to improve Figure 4 .. You have to separate EPS and AGS, and your both of X axis is the same value, don’t need to use 2 ” X axis”, try to use maybe colours to make it more clear
- Line 209-212 “SEM images were used to characterize the surface morphologies of the AGS before and after exposure to different NPs, and an energy-dispersive X-ray (EDX, FEI Quanta 200FEG, USA) was equipped for elemental analysis. The pretreatment procedure of AGS samples for SEM observation was described in Supplementary Information” – I could not find any SEM images nor in the main document neither in Supplementary material please add these images this will really improve your manuscript. As well could you explain which procedure in Supplimentary material describes pretreatment procedure of AGS samples for SEM?
Author Response

(The authors gave the same response as above.)

Reviewer 3 Report
The research is interesting and thoughtfully executed. Clearly the authors planned to cover all the bases of cause and effect, considering ionic effects as well as colloidal, and different responses to these challenges. The paper is hard to read though, with distractions of errors of spelling and grammar. It really needs a thorough going over and resubmission. Nevertheless the foundations are really strong and the paper should be published. My suggestion is that the paper be better prepared and we do it again. Unfortunately the requested turn-around time for review is not sufficient for detailed editing.
There is a lot of great use of the literature to support the observations made by this result. It is not clear from descriptions of agreement with papers where the contribution is for this research. I am sure there is more novelty than is implied by remarking that this result agrees with other papers. Whilst being confirmatory is important, it would be useful to know how this research takes that existing knowledge further (such applying to a different system or conditions).
I think given the range of angles explored in the paper, it is warranted to present a speculative diagrammatic model of what is going on.
Some examples of editing required...
Line 11 first word the to The.
Line 11. NPs in full on first use, ie ….nanoparticles (NPs)…
Line 13 EPS and AGS in full on first use.
Line 99. Claiming something is the first study is a bit hazardous, and not helpful.
125 CuO NPs a bit smaller….important?
Line 144 ropriately?
Line 158 IC50 – 50% inhibition only reached with CuO NPs so not really interpolation. Extrapolation I guess. Can say that IC50 is above a certain concentration where you don’t actually reach it.
Line 225 and Figure 1. Nice result. Are the error bars based on 3 different bottles at each data point, or based on the analytical variability?
Line 244 and throughout. Use of too many significant figures. Really two sig figs is enough, though three is ok for example Line 257 214.34 mg/L could be 214.
Line 249. typea (to types).
Figure 2. Legend needs fixing.
Line 283. Define more TB and LB EPS fractions.
Line 316-318. The data does not show a significant decline initially. Its possible that there is, but the data is too variable in this case.
Author Response

(The authors gave the same response as above.)

Reviewer 4 Report
After reviewing the manuscript, which is generally devoted to the current topic, I can draw the following conclusions.
In general, the article has a double impression: the authors conducted a lot of research, including biochemical analyzes, fish hybridization, but from the very beginning there is no clear meaning and clear idea of the research. However, the text is difficult to read, so I would advise the authors to restructure the manuscript, separating the results and discussion into separate chapters. In this form, I have not yet recommended the text for publication.
To my opinion, the conclusions about the mechanisms of toxicity given by the authors look unconvincing. For example, the difference in experiments based on data from Relative ROS production and LDH release looks unsignificant. Many conclusions are not supported by the results. For example, line 332 “The above data indicated that EPS acted as a protective barrier by keeping TiO2 and ZnO NPs distant from the cells”.
On the Figure 4. It is not clear the correlation of active archaea-bacteria in the biofilm, the processes are multidirectional. At the same time, since control data without nanoparticles are not presented, it is difficult to draw conclusions. In the “protain” there is a mistake.
When copper is added, polysaccharides are not formed at all, and, according to many literature data, it is the most toxic. Why, when adding copper nanoparticles, there is a high content of DNA in the biofilm, and in the case of titanium, which is generally less toxic, there is almost no DNA. The protein in the biofilm rather indicates the presence of cells in the matrix. Small differences in protein content indicate either an insignificant effect of nanoparticles or a large error in the method, but not their toxic effect.
The methods mention FTIR, but the results and discussion do not provide data.
Did the authors look at the sizes of nanoparticles in biomass. Are they there and what size? Most likely large and sticky, so reasoning about the toxicity of the dose / effect is ineffective.
With a high probability, the added nanoparticles in the medium at neutral pH values are unstable, probably hydrate and bind to polysaccharides and cell walls. The results given by the authors with the addition of a stabilizer show its insignificant effect. Why then it was added and given these data. It may be necessary to check different types of stabilizers. While it is not clear about the shape of nanoparticles in solution at different pH values, nothing can be said about the mechanism of toxicity and no significant conclusions can be drawn about the role of EPS. Of course, EPS protects the cell from toxic factors, but this has long been known to everyone. Based on the foregoing, I rely on the opinion of the editor about the possibility of publishing this article. My opinion is not at this stage it should be rejected or held a major revision.
Author Response

(The authors gave the same response as above.)

Round 2
Reviewer 2 Report
- Figure2 on some bars errors are missing. I can't understand what the minus sign means above the bars, there is no explanation of what the asterisks mean.
- Under all figures explanation what means asterisks means are missing.
Author Response
- We are terribly sorry for the carelessness on the graph-exporting, and we have exported a new one, please see the revised version in the attachment.
- As other reviewer has raised an advice that "the asterisks explanation could be included in the text rather than under the figure as it makes the name to be too long", we have deleted all the similar description in all the figures and described the asterisks explanation in Section 2.6.
Reviewer 3 Report
I am travelling without computer so might have missed minor errors of grammar, but well done and thank you for your full consideration of co..ents.
Author Response
We are truly grateful for your advice and encouragement, your support is really important to our work.
Reviewer 4 Report
After reviewing the revised version of the manuscript, I believe that it has become much better and more understandable. However, a number of questions remain.
Fig 4 Why do the authors talk about active bacteria in the biofilm? How is their activity measured?
It would be interesting to use the control of cells lacking EPS, then we can talk about the role of EPS in protection. Did the authors measure the EPS thickness in granules as compared to bacterial cells? In my opinion, this should be the main factor of protection. At the same time, it is likely that titanium nanoparticles interacting with EPS are completely deactivated on them, probably copper and zinc, too. On a very dense EPS, the release of metal ions (copper, zinc) is possible under aerobic conditions, but they will probably immediately interact with the EPS. Figure S5 clearly shows that the metals are on the periphery. It seems to me that the topic of EPS protection has not been fully explored (first of all, the mechanisms of toxic effects are not fully understood) and I would advise the authors to change the title of the article to a more neutral one. In this work, in fact, the authors conducted a study of the physiological and biochemical parameters of biomass under the influence of particles.
To discuss the role of particles, a TEM analysis of a sample with particles and individual particles can be done to substantiate whether particles accumulate in a biofilm. If the Ti nanoparticles are to be sufficiently stable, then the Cu and Zn particles are likely to be more prone to oxidation, which is the reason for their toxic effect. This can be confirmed by the FTIR data, where large titanium particles were more visible. The author should also have analyzed the TEM particles of all types in a cell-free solution. Perhaps their agglomeration occurs with a changed ionic strength of the solution and pH values. I have no doubt that agglomeration occurs in the EPS. It would also be good to do cell microtomy after exposure to nanoparticles, this would make it possible to understand whether there are metals inside the cells or whether they are all in the EPS outside the cell. EDÐ¥ analysis of cell sections would make it possible to determine, based on the content of phosphorus, potassium and other vital elements in them, their physiological state, as well as the integrity of the cell wall. It may be worthwhile for the authors to conduct an experiment to increase the productivity of EPS using some modifying additives in the environment in order to assess the toxicity of nanoparticles under new conditions.
I understand that these are additional possibly long studies and would advise the authors to change the aspect of the manuscript to describe the toxic effect and draw conclusions about the role of EPS in the next work.
